# Experimental Study on Critical Parameters Degradation of Nano PDSOI MOSFET under TDDB Stress

**DOI:** 10.3390/mi14081504

**Published:** 2023-07-27

**Authors:** Tianzhi Gao, Jianye Yang, Hongxia Liu, Yong Lu, Changjun Liu

**Affiliations:** Key Laboratory for Wide Band Gap Semiconductor Materials and Devices of Education, School of Microelectronics, Xidian University, Xi’an 710071, China; 21111212784@stu.xidian.edu.cn (T.G.); jyyang_1997@stu.xidian.edu.cn (J.Y.); yonglu_1999@stu.xidian.edu.cn (Y.L.); liuchj@stu.xidian.edu.cn (C.L.)

**Keywords:** time-dependent dielectric breakdown, critical parameters, threshold voltage shift, gate oxide, partially depleted silicon-on-insulator

## Abstract

In today’s digital circuits, Si-based MOS devices have become the most widely used technology in medical, military, aerospace, and aviation due to their advantages of mature technology, high performance, and low cost. With the continuous integration of transistors, the characteristic size of MOSFETs is shrinking. Time-dependent dielectric electrical breakdown (TDDB) is still a key reliability problem of MOSFETs in recent years. Many researchers focus on the TDDB life of advanced devices and the mechanism of oxide damage, ignoring the impact of the TDDB effect on device parameters. Therefore, in this paper, the critical parameters of partially depleted silicon-on-insulator (PDSOI) under time-dependent dielectric electrical breakdown (TDDB) stress are studied. By applying the TDDB acceleration stress experiment, we obtained the degradation of the devices’ critical parameters including transfer characteristic curves, threshold voltage, off-state leakage current, and the TDDB lifetime. The results show that TDDB acceleration stress will lead to the accumulation of negative charge in the gate oxide. The negative charge affects the electric field distribution. The transfer curves of the devices are positively shifted, as is the threshold voltage. Comparing the experimental data of I/O and Core devices, we can conclude that the ultra-thin gate oxide device’s electrical characteristics are barely affected by the TDDB stress, while the opposite is true for a thick-gate oxide device.

## 1. Introduction

Silicon-on-insulator (SOI) technology has always been the focus of research in the field of microelectronics. Compared with bulk silicon devices, due to their advantages of full medium isolation and high-temperature resistance, SOI devices are widely used in [1,2]. With the shrinking size of the integrated circuit and the thinning gate oxide thickness of MOSFETs, time-dependent dielectric breakdown (TDDB) remains a key reliability concern in MOSFETs in recent years [3,4,5]. Many researchers focus on studying the TDDB lifetime of advanced devices and the mechanism of oxide damage [6,7,8].

According to previous research, TDDB stress can lead to trap charge accumulation in MOSFET gate oxide. Researchers have proposed numerous models to interpret the degradation mechanism of TDDB. The typical models are the impact ionization and recombination model and the thermodynamic breakdown model, which explain the mechanism of charge generation in the oxide layer [9]. When trap density reaches a certain threshold, a conductive path will form in the gate oxide [10]. The oxide charge will reduce the mobility of MOSFET channel carriers, affecting the formation of the reverse layer and the gate’s ability to control it. According to this mechanism, the oxide charge can affect the MOSFET properties. The MOSFET *I-V* characteristic curves, threshold voltage, and transconductance will be shifted during the TDDB stress. When the TDDB effect is combined with additional reliability effects such as the radiation effect and HCI (hot carrier induce) effect, the changes in device characteristics will be further complicated [11,12,13,14]. In recent research, different TDDB mechanisms were discovered in modern-generation semiconductor devices [15,16]. However, the physical mechanism leading to the TDDB effect is still uncertain. There appears to be limited research on MOSFET parameter degradation while applying TDDB acceleration stress. The degradation data of the critical parameters would be an excellent way to understand the TDDB mechanism of the MOSFET.

In addition, the experimental samples used in this paper include the I/O devices and the Core devices, which are the thick-gate oxide devices and ultra-thin gate oxide devices. It is helpful to investigate the relationship between the thickness of different gate oxides and TDDB stress damage.

In this work, we experiment with two different types of devices to investigate the TDDB degradation pattern of the MOSFET. Sensitive parameter degradation due to TDDB acceleration stress in a 130 nm PDSOI device, including transfer characteristics, output characteristics, threshold voltage, and transconductance, were investigated.

## 2. Materials and Methods

The test example is PDSOI NMOSFET manufactured by the 130 nm process line of Hua Hong Semiconductor Grace Semiconductor Manufacturing Corporation (Shanghai, China). The devices have a nominal operating voltage of Vdd = 3.3 V (I/O device) and Vdd = 1.2 V (Core device). The top Si film has a thickness of 100 nm and the buried oxide has a thickness of 145 nm. The I/O and Core devices were selected for this experiment. The Gate oxide material of the I/O and Core devices is SiO_2_. The largest difference between the two types of devices is that the I/O and Core device’s oxide gate thickness are 7 nm and 2 nm, respectively. Table 1 shows the detailed data for the two types of devices. DIP (dual in-line package) 24 was used to improve test reliability. In order to suppress the floating body effect, the I/O and Core devices are both T-gate structures. The layout and structure of the device is shown in Figure 1.

The TDDB acceleration stress was conducted by an Agilent B1500A semiconductor parameter analyzer (Canada, Americas) [17]. In the TDDB acceleration stress experiment, we needed to apply high stress to the sample for a long time. Therefore, we used constant voltage stress (CVS) [18,19,20,21] on the devices to monitor the changes in parameters and used a custom test box to connect the B1500A’s cable to the devices to guarantee the stability of the test. This test selected 15 samples manufactured in the same batch consisting of the same components to test the I/O devices and Core devices on different samples. Before the experiment, we needed to test the initial characteristics and the gate intrinsic breakdown voltage of the device. The initial characteristics can be used to confirm that the device is functioning properly, while the gate intrinsic breakdown voltage can be used to determine the applied voltage for TDDB acceleration stress experiments. Therefore, we first used two sets of samples as pre-tests, and we used the slope method to measure the initial breakdown characteristic curve of the device. The initial breakdown characteristic is shown in Figure 2. The results show that the breakdown voltage of the I/O device with a working voltage of 3.3 V is 8.5 V, and the breakdown voltage of the Core MOS device with a working voltage of 1.2 V is 3.7 V. The gate of the device was at a higher voltage than the normal operating voltage. We used 80% of the intrinsic breakdown voltage as the stress voltage. Therefore, the gate stress voltage of the I/O device is 7.0 V and that of the Core device is 3.0 V. The source, drain, and substrate contact were connected to 0 V (grounded). The TDDB acceleration stress was applied to the device until oxide breakdown was observed. During the TDDB test, the parameters were measured at intervals so that we could obtain the degradation of the parameter during the stress. By using the B1500A semiconductor tester, we were able to obtain the electrical parameters of the device automatically. The following device curves and parameters were measured to monitor device degradation under stress, such as the transfer characteristic curve, output characteristic curve, threshold voltage, and transconductance.

## 3. Results and Discussion

To analyze the key factors affecting the TDDB effect, we will analyze the degradation of the device I-V characteristics, critical parameters, and gate current curves during the TDDB acceleration stress. The specific analysis process is as follows.

The total stress testing time of the sample is 4500 s, and the temperature is fixed at room temperature. The constant voltage method is used to measure the core device at an input voltage of 3 V, and the core device at an input voltage of 7 V. The transfer and output characteristic curves of the device are tested after applying stress for 0 s, 1500 s, 3000 s, and 4500 s. The specific testing steps are as follows. Firstly, a constant voltage stress will be applied to the sample. The transfer characteristic and the output characteristic will be extracted by a computer program. Secondly, the critical parameters of the device, including the threshold voltage and the transconductance, will be extracted from the data of the constant voltage stress experiment. Finally, the gate current curve degradation of the devices during the TDDB will be analyzed. By comparing the data of I/O and core devices, we found the effect of TDDB stress on I-V characteristics, key parameters, and the effect of TDDB degradation on oxide thickness.

### 3.1. TDDB Degradation of I-V Characteristic

Usually, we consider the TDDB process to be divided into two phases based on previous studies [22,23,24]. The first phase of the TDDB process is the defect accumulation phase, in which the trap charge continues to accumulate in the oxide layer as the stress time increases. The second phase is the breakdown phase. When the trap charge’s density accumulated in the first phase reaches a critical value, a complete conductive path will be formed in the MOSFET device’s gate oxide [25]. This results in the breakdown of the oxide layer. In this experiment, we will focus on the first phase of TDDB breakdown. Based on the breakdown theory described above, traps that accumulate in the gate oxide of a MOSFET must affect its performance. Therefore, the I-V characteristic and the critical parameters of the device are used to characterize the performance degradation of the device.

Figure 3 shows the transfer characteristic curve of I/O devices under accelerated stress at 7.0 V TDDB at 25 °C. It can be observed that under the action of TDDB acceleration stress, the overall transmission characteristic curve of the device shows a positive deviation trend. Among them, the speed is faster in the first 1000 s, and as the stress time continues to increase, the amplitude of the curve change gradually decreases, ultimately reaching saturation. To further understand the influence of the TDDB’s effect on device electrical parameters, a quantitative analysis of parameter curves was conducted. In Figure 3, the drain current at Vg = 3.3 V decreases from 260 μA to 200 μA in the first 1000 s, while it decreases from 170 μA to 157 μA in the last 3000 s. Figure 4 shows the degradation of the transfer characteristics of the Core device under the TDDB acceleration stress. Compared to the transfer characteristic curves of the I/O device, the curves of the Core device show much less variation even when the equivalent oxide electric field is higher than the former. The difference value of Id between t = 0 s and t = 7500 s in the Core device is below the nanoampere level.

Figure 5 shows the variation in the output characteristic curves of the I/O device under 7.0 V TDDB acceleration stress. Like the transfer characteristic curves, the I/O device curves change rapidly in the first 1000 s of stress. The output current Id, which at Vd = 3.3 V and Vg = 3.3 V, reduced from 3.16 mA to 2.24 mA, which is almost a 50% reduction in the first 1000 s. However, the output current decreases only from 1.78 mA to 1.61 mA between 3000 and 4500 s. As the stress time increases, the variation in the curve saturates with time. The Core device also behaves differently from the I/O device at the output characteristic curve, as shown in Figure 6. The change in current from 0 to 4500 s is very small. The percentage change is less than 1%. The variation is too small to be observed even in the enlarged picture. However, as with I/O devices, the value of the current is still decreasing with time.

The phenomenon of I-V characteristic drift was also found in previous studies. Some researchers performed a TDDB experiment with high-k/metal dielectric devices [26]. The variation in the transfer characteristic curves and output characteristic curves of the devices in their experiments are similar to those in this paper. Therefore, the main reason for the degradation of the device transfer characteristics and the output characteristics is the gate oxide negative charge induced by the TDDB acceleration stress. The negative charge in the oxide layer can induce the positive charge in the channel. The positive charge in the channel side will prevent the formation of an inverse layer, leading to a shift to the right in the transfer characteristic curve and a reduction in the drain current in the output characteristic curve. Comparing I/O and Core device I-V characteristic degradation, we can find that the I-V characteristic degradation patterns are similar for both types of devices. However, the data variation in the Core device is much smaller than that of the I/O device. In the Core device, the gate oxide layer is thin enough that the voltage stress can hardly break the covalent bond, so the channel charge will accumulate less than the I/O device. The biggest difference between the I/O device and the Core device is that the gate oxide in the latter is much thinner. The thicker the gate oxide layer, the higher the stress-induced trap concentration. Hence, the I-V characteristic degradation of the Core device is much smaller.

### 3.2. TDDB Degradation of Critical Parameters

The transistor threshold voltage is the gate voltage at which the device begins to switch on. Moreover, the transconductance is a manifestation of the ability of the gate voltage to control the drain current. These two parameters are critical parameters of MOSFET and can reflect the gate oxide quality and performance of MOSFET. Figure 7 shows the threshold voltage and maximum transconductance degradation of an I/O device under the TDDB stress. It can be observed that both the threshold voltage and the transconductance of the I/O device are significantly affected by the stress. In particular, at early times, the threshold voltage shifts from 539 mV, corresponding to t = 0 s to 1020 mV, corresponding to t = 1500 s. Like the transfer characteristic curves of I/O devices, the amount of variation in the threshold voltage also decreases with stress time. The maximum transconductance of the I/O device is shifted negatively, with the same trend as the threshold voltage. Since the transconductance and threshold voltage are closely related to the oxide trap density, it can be inferred that the TDDB acceleration stress leads to an increase in the oxide trap density. However, the two parameters of the MOSFET are almost unchanged under the stress in the Core device shown in Figure 8.

From the threshold voltage diagram and the transconductance diagram above, it can be seen that the TDDB acceleration stress leads to significant degradation of the critical parameters. The stress-induced negative charge in the oxide layer leads to the induced positive charge formed in the MOSFET channel. The positive charge accumulation in the channel is equivalent to increasing the P-type doping concentration (*N_a_*) near the channel. According to the working principle of NMOSFET, the higher the substrate *N_a_*, the higher the threshold voltage. When the stress-induced charge leads to an increase in the concentration of *N_a_* in the vicinity of the channel, a larger gate voltage will be required for the formation of the inversion layer. The result is a positive shift of the threshold voltage. The stress-induced charge near the Si-SiO_2_ interface also affects the mobility of the carrier. In general, the transconductance of a MOSFET is proportional to the carrier mobility. So, we can observe in Figure 7b that the transconductance decreases with stress time. Figure 8a,b shows the critical parameter degradation for the Core device. It can be observed that the relative changes of the threshold voltage and the maximum transconductance are less than 1% and can be considered constant. This phenomenon indicates that the TDDB acceleration stress does not produce a large number of traps in the ultra-thin gate oxide to affect the performance of the MOSFET.

### 3.3. The Gate Current Curves during the TDDB Acceleration Stress

Figure 9 and Figure 10 show the typical gate current curves during the TDDB acceleration stress for the I/O and Core devices. It is clear that the gate current of the I/O device continuously decreases until the breakdown occurs. The gate current of the Core device, however, remains nearly constant during the stress time. We also observed that there is a soft breakdown [27] in the Core device, as shown in Figure 10a. The *I-V* characteristics, threshold voltage, and transconductance remain the same before and after the soft breakdown in the Core device. In JY Shen’s study of 2 nm and 5.6 nm thick-gate oxide NMOS capacitors, there are several soft breakdown events in the 2 nm ultra-thin-gate oxide [14]. When the soft breakdown occurs, the gate current jumps up. Although it is not very obvious in the picture, there is still a soft breakdown point in the *I_g_-t* characteristic. Soft breakdown was also found in Kim’s study [28].

Based on the above phenomena, we can believe that the TDDB stress can generate traps in MOSFET oxides. The I/O devices used in the experiment have 7 nm oxide and the Core devices have 2 nm oxide. The thicker gate oxide in the I/O device leads to more traps during the TDDB stress. It is the oxide trap that leads to the shift of the *I-V* characteristics, threshold voltage, and transconductance. In the thick oxide I/O devices, the continuous application of stress leads to the accumulation of oxide trap charges. There is plenty of space in the thick-gate oxide, so the trap charge can generate an inverse electric field to counteract the applied electric field. The result is a drop in the current shown in Figure 9. We also note that the current drop is significant only in the early part of the TDDB acceleration stress application. The variation in the current becomes less and less, which corresponds to the previous experimental data for the electrical parameters. In an ultra-thin gate oxide device, the Core device, TDDB stress will generate far fewer oxide traps, such that its *I-V* characteristic curves and critical parameters are nearly constant. The soft breakdown shown in Figure 10a is caused by the conductive path induced by the trap charge. The gate oxide thickness of the Core device is thinner than that of the I/O device, so the trap density threshold for breakdown is lower than that of the I/O device. The above causes the soft breakdown in the Core device, which is recoverable after removing the TDDB acceleration stress. The rap accumulation rate in the ultra-thin-gate oxide device is much slower than that in the thick-gate oxide device. Hence, the *I-V* characteristics and critical parameters of the Core device will barely change during the TDDB acceleration stress. When the charge density reaches a certain density, the gate oxide of the Core device breaks down. In addition, the interface roughness of the oxide layer has a large impact on the dielectric properties in ultra-thin-gate oxide devices. For the 2 nm gate oxide devices used in this paper, the thinnest part of the gate oxide can be less than 1.5 nm or even less. This is also the reason why soft breakdown occurs in Core devices.

## 4. Conclusions

With the development of technology, SOI devices have been proven to be nanoscale device structures. Time-dependent dielectric electrical breakdown (TDDB) has become one of the important degradation mechanisms of PDSOI devices. However, most of the analysis on the TDDB effect of MOS devices focuses on the breakdown time, and little attention is paid to the degree to which the key parameters of the device are affected by the TDDB effect. Therefore, this article provides a detailed study of the TDDB effect on the main characteristic parameters of PDSOI devices, such as threshold voltage and transconductance. In this paper, we investigate the critical parameter degradation of I/O and Core PDSOI devices under TDDB acceleration stress. The degradation of the I-V characteristics and critical parameters of the devices under TDDB acceleration stress was tested. It was found that the TDDB acceleration stress leads to a positive shift in the transfer characteristic curve and a decreasing output current with time in both the I/O and Core devices. However, critical parameters such as the threshold voltage and the degradation of the transconductance during stress show differences in I/O devices and Core devices. In the I/O device, the threshold voltage and the transconductance change rapidly under stress, while in the Core device, they remain constant due to the difference in oxide thickness. By measuring the TDDB curves of the I/O device and the Core device, we find that the gate current of the thick-gate oxide device decreases with time under stress. However, in ultra-thin-gate oxide devices, there is soft breakdown.

## Figures and Tables

**Figure 1 micromachines-14-01504-f001:**
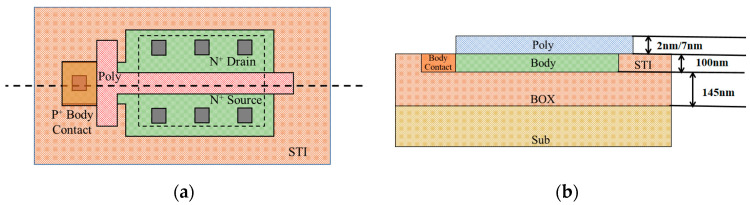
Layout of T-gate transistor of PDSOI used in our study (**a**) top view, (**b**) sectional view.

**Figure 2 micromachines-14-01504-f002:**
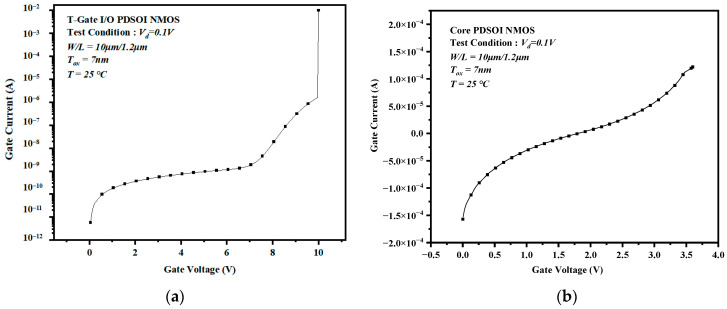
Initial breakdown characteristic curves of the (**a**) I/O devices and (**b**) Core devices under the TDDB stress.

**Figure 3 micromachines-14-01504-f003:**
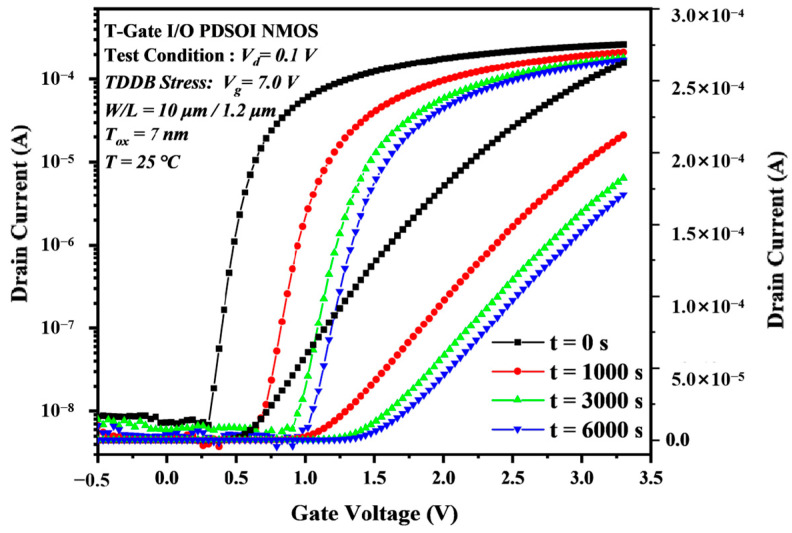
Degradation of the transfer characteristic curves of the I/O devices under the TDDB stress.

**Figure 4 micromachines-14-01504-f004:**
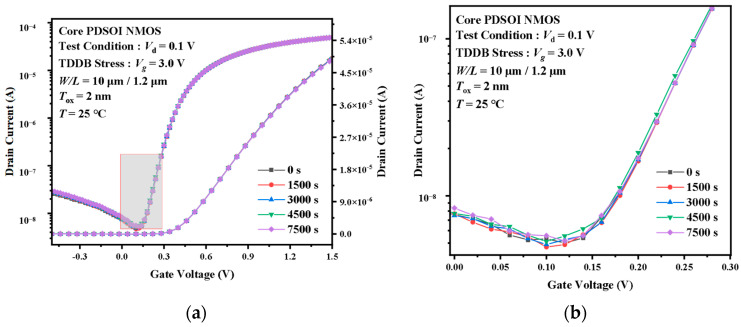
Degradation of the transfer characteristic curves of the Core devices under the TDDB stress. (**a**) Complete, (**b**) partially enlarged.

**Figure 5 micromachines-14-01504-f005:**
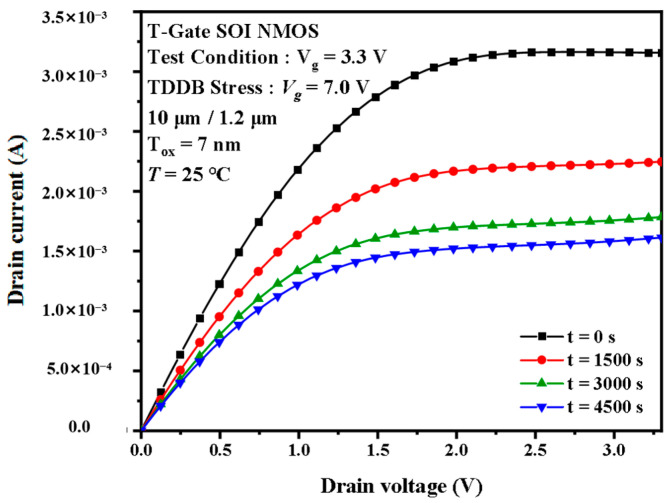
Degradation of the output characteristic curves of the I/O devices under the TDDB stress.

**Figure 6 micromachines-14-01504-f006:**
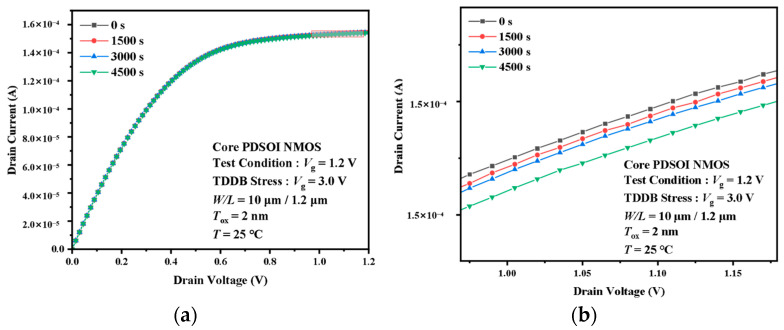
Degradation of the output characteristic curves of the Core devices under the TDDB stress. (**a**) Complete, (**b**) partially enlarged.

**Figure 7 micromachines-14-01504-f007:**
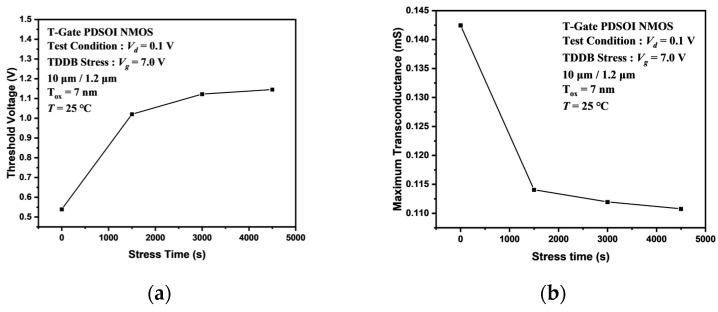
Degradation of the critical parameters of the I/O devices under the TDDB stress. (**a**) Threshold voltage, (**b**) maximum transconductance.

**Figure 8 micromachines-14-01504-f008:**
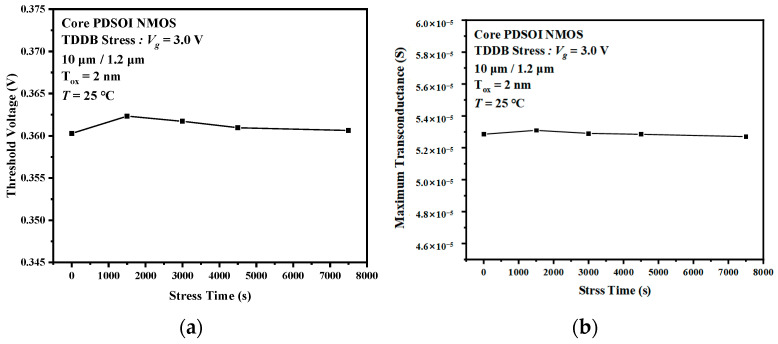
Degradation of the critical parameters of the Core devices under the TDDB stress. (**a**) Threshold voltage, (**b**) maximum transconductance.

**Figure 9 micromachines-14-01504-f009:**
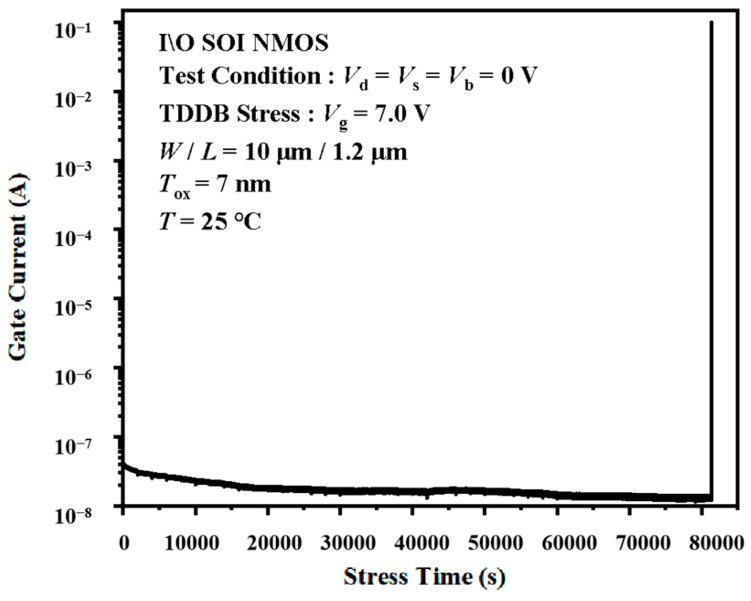
Gate current during the TDDB test of I/O device.

**Figure 10 micromachines-14-01504-f010:**
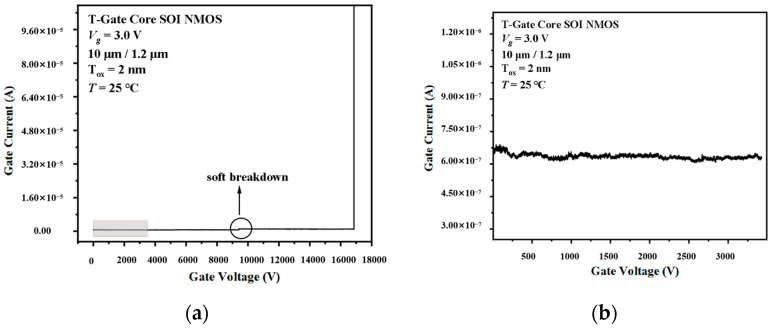
Gate current during the TDDB test of Core device. (**a**) Complete, (**b**) partially enlarged.

**Table 1 micromachines-14-01504-t001:** The detailed information of Core and I/O devices in the experiment.

Device	Operating Voltage	Gate Oxide Thickness	Width-Length Ratios (W/L)	Body Contact
Core	1.2 V	2 nm	10 μm/1.2 μm	Yes
I/O	3.3 V	7 nm	10 μm/1.2 μm	Yes

## Data Availability

Not applicable.

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
