# Peer review of "Experimental Study on Critical Parameters Degradation of Nano PDSOI MOSFET under TDDB Stress"

_micromachines, 2023, doi:10.3390/mi14081504_

Round 1

Reviewer 1 Report (New Reviewer)

Comment 1: What do the terms I/O device and core device mean, and why are they not shown in Fig. 1? If the only difference between these devices is the gate oxide thickness, why are they called I/O and core devices? “The main difference between the two types of devices is that the gate oxide thickness of I/O and core devices is 7 nm and 2 nm, respectively.”

Comment 2: This work seems to be a theoretical study, not an experimental one.

Comment 3: The geometrical parameters of the devices should be specified in Fig.1, such as the top Si film thickness, the buried oxide thickness, and the embedded oxide layer thickness.

Comment 4: What does the author mean by “DIP 24 package”?

Comment 5: The authors seem confused about the correct thickness of I/O and core devices. For example, see 1) the gate oxide in Fig.2(b) should be 2 nm; 2) “The constant voltage method is used to measure the core device at an input voltage of 3V, and the I/O device at an input voltage of 7V.” This issue should be checked throughout the manuscript.

Comment 6: I think the results of this research are contrary to the common knowledge that thicker gate oxide leads to higher reliability. How can the authors explain this discrepancy? Is there any logical reason behind it?

The text should be checked again for grammatical details and prepositions.

Author Response

Thank you for your comment.The suggestions you provided are very helpful to us, helping us with our future implementation process. Thank you for your suggestion. We are very sorry for the improper description in the original manuscript due to our poor English writing skills. The manuscript has been revised, though there might still be some mistakes we haven’t discovered due to the time limit. Once again, we are very sorry for the inconvenience of reading due to our poor English writing skills

Reviewer 2 Report (New Reviewer)

General:
There is some repetitions in the text, e.g. for film thickness, buried oxide thickness, measurement voltages

Line 80:
What does the custom test box do? Is it just for connecting or does it contain other circuitry or logic?

Line 134-137:
"off-state current .. shifts to the right .." should probably be reformulated
"However, the transfer characteristic curve is positively shifted ... and faster for the first 1000s" is hard to understand for me what exactly the authors want to say

Figure 4b:
Why does the drain current seem to increase towards Vg = 0V?

There is a lot of double/missing spaces and punctuation, some word repetitions and spelling mistakes, which can be fixed

Author Response

Thank you for your comment. The suggestions you provided are very helpful to us, helping us with our future implementation process. Thank you for your suggestion. We are very sorry for the improper description in the original manuscript due to our poor English writing skills. The manuscript has been revised, though there might still be some mistakes we haven’t discovered due to the time limit. Once again, we are very sorry for the inconvenience of reading due to our poor English writing skills

Round 2

Reviewer 1 Report (New Reviewer)

The authors have answered all the comments. At the present time, the manuscript can be published in the Journal.

No Comment

This manuscript is a resubmission of an earlier submission. The following is a list of the peer review reports and author responses from that submission.

Round 1

Reviewer 1 Report

Thanks for promoting your English quality in your revised article. As I have mentioned in the previous comments, the TDDB stress experiment should cover the temperature effect to extract the lifetime and the activation energy. Thus, the tested samples with package level are numerous, not only couple number. In your proposed results, I didn’t see the special failure mechanism(s) related to TDDB stress. It seems that your proposed data is not interesting to the readers. In the industrial test, the stress time is up to 1week or more. Thus, few people use the wafer-level probing in TDDB test because the probe needles will move out from the tested pads, causing the test error. Furthermore, the device format is with PDSOI. In the published literature, the FDSOI device in TDDB stress also was studied more. By the way, the gate dielectric is silicon dioxide with 2nm thickness in your article. The gate leakage and the uniformity of thickness of gate oxide on wafer are very important in gate oxide growth. In Fig. 9(b), the initial gate leakage at VG=3V is, at least, 5A/cm2, which is too huge to be allowed in IC design. You should propose IG-VG characteristics for core and I/O tested devices in your article.

  Finally, in 130nm process manufacturing, the gate dielectric is basically with nitrogen treatment, not pure silicon dioxide. If it is true, the degradation mechanisms for pure SiO2 and SiON/SiO2 are different. Unfortunately, these differences have been investigated more.

The English quality is fine.

Reviewer 2 Report

Before publication, there are corrections to be made, and in the case of figure 9 it is mandatory.

Minor editing of English language required.

Reviewer 3 Report

The authors present an experimental study on the degradation of PD SOI MOSFET. This work might be interesting for readers, but it needs extensive improvements.

- Authors should cite recent publications and review papers on TDDB. Many works regarding TDDB have been published.

- Authors claim that they study 130 nm PDSOI device. However, according to Table 1. it is a 1.2-um gate-length device. Describe the studied device clearly.

- The measurement procedure and used box set should be clearly described.

- What is the gate oxide breakdown voltage value of the studied device? What is the gate oxide material in the gate stack? Is it SiO, SiON,..? Details of the gate stack composition should be presented.

- In general, more data is needed to draw any conclusions. TDDB is a statistical effect. The authors show measurement results of only one device with different oxide thicknesses.

A lot of work on TDDB has been published already. Thick gate oxides have been studied for many years. What is the novelty of this paper? I don't see it. In my opinion, authors should focus on studies of thin oxide layers and relevant phenomena.

English language improvement is needed. There are errors in sentences.